# Introduction of a Novel Sequential Approach to the Ponte Osteotomy to Minimize Spinal Canal Exposure

**DOI:** 10.3390/children10030470

**Published:** 2023-02-27

**Authors:** Ian Hollyer, Taylor Renee Johnson, Stephanie Tieu Kha, Cameron Foreman, Vivian Ho, Christian Klemt, Calvin K. Chan, John Schoeneman Vorhies

**Affiliations:** 1Department of Orthopedic Surgery, Lucile Packard Children’s Hospital at Stanford, Stanford, CA 94304, USA; 2Department of Orthopedic Surgery, Stanford University, Stanford, CA 94305, USA

**Keywords:** adolescent idiopathic scoliosis, Ponte osteotomy, pediatric

## Abstract

Ponte osteotomy is an increasingly popular technique for multiplanar correction of adolescent idiopathic scoliosis. Prior cadaveric studies have suggested that sequential posterior spinal releases increase spinal flexibility. Here we introduce a novel technique involving a sequential approach to the Ponte osteotomy that minimizes spinal canal exposure. One fresh-frozen adult human cadaveric thoracic spine specimen with 4 cm of ribs was divided into three sections (T1–T5, T6–T9, T10–L1) and mounted for biomechanical testing. Each segment was loaded with five Newton meters under four conditions: baseline inferior facetectomy with supra/interspinous ligament release, superior articular process (SAP) osteotomy in situ, spinous process (SP) osteotomy in situ, and complete posterior column osteotomy with SP/SAP excision and ligamentum flavum release (PCO). Compared to baseline, in situ SAP osteotomy alone provided 3.5%, 7.6%, and 7.2% increase in flexion/extension, lateral bending, and axial rotation, respectively. In situ SP osteotomy increased flexion/extension, lateral bending, and axial rotation by 15%, 18%, and 10.3%, respectively. PCO increased flexion/extension, lateral bending, and axial rotation by 19.6%, 28.3%, and 12.2%, respectively. Our report introduces a novel approach where incremental increases in range of motion can be achieved with minimal spinal canal exposure and demonstrates feasibility in a cadaveric model.

## 1. Introduction

Adolescent idiopathic scoliosis (AIS) affects approximately 2–4% of adolescents and is a complex three-dimensional deformity of the spine characterized by abnormalities in the coronal, sagittal, and axial planes [1,2]. Treatment depends on curve magnitude and the patient’s skeletal maturity, ranging from observation or bracing to surgery [3,4]. In general, patients with major Cobb angles greater than 50 are indicated for surgery in the form of spinal fusion [5]. Surgery aims to halt curvature progression, improve sagittal and coronal balance, reduce short-term and long-term complications associated with AIS, and improve patient appearance [4,6].

Historically, surgeons used a combination of anterior and posterior releases of the spine to increase spinal flexibility and improve deformity correction before fusion, as anterior discectomy was considered necessary in patients with stiff curves [6]. In recent years, however, biomechanical advances in pedicle-based instrumentation have dramatically increased the correction force that surgeons can apply to the spinal column for deformity correction and have lessened the need for combined releases [7]. Due to the morbidity associated with anterior approaches, many surgeons now advocate for posterior-only approaches for spinal fusion, even for patients with large curves [8,9]. 

Despite advances in powerful instrumentation, osteotomies of the spine are still sometimes necessary to correct more significant deformities [6]. One such osteotomy is the Ponte-type osteotomy, a posterior-column-based technique that can be performed immediately prior to posterior spinal fusion with pedicle screw implants [10,11,12,13,14]. Although the Ponte osteotomy was initially described to treat sagittal deformities, it has since been modified and is now a widespread technique for coronal, rotational, and sagittal plane correction in major thoracic curves. The Ponte osteotomy has also been described for use in thoracolumbar and lumbar curve correction [15]. In this technical note, we will focus on thoracic correction.

Ponte first described what is now known as the Ponte osteotomy procedure for correcting the sagittal-plane deformity associated with Scheuermann kyphosis in 1987 [16]. The traditional Ponte osteotomy, as described by Ponte, is as follows (see Figure 1).

“Spinous processes are resected at their base to allow better visualization of the bony parts to be removed… An angled double-action rongeur and/or a Kerrison is used to perform the bony resections. Complete facetectomies and wide inferior and superior laminectomies are performed at every intersegmental level… A generous resection of facet joints and laminae, in severe deformities as far as the pedicles, is an essential step of the osteotomy and the technique… The ligamentum flavum is removed entirely at all levels.” [17].

As seen from Ponte’s description of the Ponte osteotomy, the complete removal of the spinous process and a wide lamina resection fully expose the spinal canal in the surgical field. This leaves the spinal cord at risk within the surgical field and may not be fully covered in bone even after compression of the posterior elements. Using rongeurs in the spinal canal also often results in epidural bleeding and theoretically increases the risk of dural tears or neurologic injury. Recent studies have also shown that Ponte osteotomies increase rates of intraoperative neuromonitoring alerts and blood loss during posterior spinal fusion [18,19,20,21]. Recently, ultrasonic bone-cutting devices have facilitated changes in osteotomy techniques to minimize exposure of the spinal canal in the surgical field. 

Here we describe a novel sequential approach and modification of the Ponte osteotomy that aims at keeping the spinal cord covered by bone in the surgical field if possible. The method only progresses to a full Ponte osteotomy with exposure of the spinal canal when the prior sequential osteotomy steps cannot achieve appropriate spinal flexibility and curve correction. Ultrasonic bone cutters allow in situ osteotomies of the superior articular process (SAP) and spinous process (SP), releasing the ligamentous tethers posteriorly with minimal bony resection. This approach allows surgeons to achieve the desired amount of spinal flexibility to facilitate deformity correction while minimizing exposure to the spinal canal. We investigated this aim using a human cadaveric non-scoliotic thoracic spinal specimen, an established biomechanical model, to study AIS surgical techniques.

## 2. Materials and Methods

### 2.1. Surgical Procedure

The Ponte osteotomy, as described above, involves the removal of the spinous process, facet joints, lamina, and ligamentum flavum, and by definition, exposes the spinal canal in the surgical field. Our novel technique involves sequential step-wise osteotomies, which we typically combine at multiple vertebral levels in vivo. If more flexibility is needed after the first round of osteotomies, then the next sequential osteotomy step is performed. It should be noted that an oscillating saw and osteotome were used to perform osteotomies on the cadaver specimen for this technical report, while in vivo, an ultrasonic scalpel was used. The sequential osteotomies, which were subjected to mechanical testing, were as follows (Figure 2).

Baseline: Supraspinous and interspinous ligaments were cut and partially excised using a rongeur. Inferior facetectomy was performed using an osteotome. The inferior articular processes were removed along with all the visible dorsal facet joint capsule.

O1: The in situ SAP osteotomy was performed using an oscillating saw. The cut began in the exposed SAP cartilage surface, and the saw was directed ventrally and cranially to avoid entering the pedicle at that level. Once this osteotomy was complete, the SAP fragment was confirmed to be detached from its vertebra but remained tethered cranially by the joint capsule and ligamentum flavum.

O2: The SP osteotomy in situ was performed using an oscillating saw to connect the left and right facetectomy cuts across the lamina. The saw was directed ventrally and cranially to terminate the cut ventrally near the superior aspect of the attachment of the ligamentum flavum at that level, resulting in partial central ligamentum flavum release. After this step, the epidural space was typically visible through a small gap in the lamina. The lamina/spinous process fragment was free from the level above but remained tethered to the level below by the ligamentum flavum. The fragment was left in place to protect the canal and act as a bone graft.

O3: A complete posterior column osteotomy was performed using a rongeur to remove the SP/lamina fragment. A Kerrison rongeur was then used to release any remaining ligamentum flavum laterally and remove the SAP fragment. After this step, the spinal canal was open, and dura and epidural fat were visible. The exposure was now the same as if a Ponte osteotomy had been initially performed. 

The primary outcomes were average degree change and percent change from baseline range of motion (ROM) under load for each sequential condition for the three cadaveric sections. 

Biomechanical testing was first performed on the three initial spinal specimen sections from the single cadaver and repeated after each successive procedure. A board-certified pediatric orthopaedic surgeon performed osteotomies to ensure appropriate technique.

### 2.2. Specimen Preparation

One fresh-frozen human cadaveric thoracic spine specimen (age 57, BMI of 18) from T1-L1 with 4 cm of ribs was divided into three sections (T1–T5, T6–T9, T10–L1). All specimens were dissected free of paraspinal musculature to include only stabilizing ligaments, bones, intervertebral discs, and 4 cm sections of ribs. All specimens were studied using radiographs to ensure no history of fractures, osseous abnormalities, osteoporosis, bridging osteophytes, or previous spine surgery. Specimens were thawed for 24 h at room temperature before testing. 

### 2.3. Biomechanical Testing and Analysis

A simVITRO robotic testing system (Cleveland Clinic, Cleveland, OH, USA) with a KR300 robot (Kuka, Ausburg, Germany) and an Omega160 6-axis load cell with an SI-2500–400 calibration (ATI, Apex, NC, USA) was used to apply the single-plane ranges of motion to the spine (Figure 3). Potting involved drilling a 3-inch wood screw placed anterior to posterior through the most inferior and superior vertebral body into the spinous process, which was confirmed with fluoroscopy. This wood screw then sat within the trough of the pot. Poly-methyl-methacrylate (PMMA) cement was added into the mold parallel to the vertebral end plate of the interior and superior vertebra. In order to mount each spine to the robot, the superior and inferior vertebrae of each specimen in PMMA blocks were rigidly fixed to custom clamps to prevent spine movement relative to the base and robot arm. Testing was conducted in all three modes of bending (flexion-extension, lateral bending, and axial rotation). Phosphate buffered saline solution was used to keep the soft tissue structures hydrated and preserve the mechanical integrity of the specimen throughout testing. Throughout the testing process, there was no evidence of loosening. Spines were mounted onto the robotic platform and initialized by determining the spatial relationships between the robot, load cell, and each vertebral segment using a Romer Absolute Arm Digitizer (Hexagon, RI, USA). Vertebral body coordinate systems definitions were established according to International Society of Biomechanics standards [22]. Joint motion and load were controlled by establishing geometric relationships to a coordinate system, and any changes in biomechanical responses were recorded. Additional information regarding the programming of the robot and coordinate system analysis has been described by Mageswaran et al. [23].

Loading conditions were performed along three primary single-plane axes (±5 Nm moment in flexion-extension, lateral bending, and axial rotation) while minimizing the load along the translational axes (Figure 3). To eliminate viscoelastic effects, the specimens were preconditioned for four cycles before measuring the fifth cycle.

## 3. Results

### 3.1. Average of the Entire Single Specimen Spine T1–T12

Under 5 Nm of torque, the T1–T12 percent change in flexion-extension, lateral bending, and axial rotation steadily increased in a stepwise fashion following initial SAP osteotomy (O1), a subsequent SP osteotomy (O2), and finally completion PCO with excision of the ligamentum flavum (O3). Final flexion-extension increased from the baseline specimen by 3.5% after O1, 15% following O2, and 19.6% following O3. The greatest percent change was in lateral bending, in which sequential osteotomies increased final ROM from the baseline specimen by 7.6% after O1, 18.3% following O2, and 28.4% following O3. Lastly, axial rotation increased by 7.2% after O1, 10.3% following O2, and 12.2% following O3 relative to the baseline specimen. In all testing conditions, the most significant percentage increase occurred following SP osteotomies (O2); flexion-extension increased by 11.5% from O1 to O2, whereas lateral bending increased by 10.7%, and axial rotation increased by 3.1% (Table 1, Figure 4).

### 3.2. Upper Thoracic Segment T1–T4 from the Single Thoracic Specimen

Flexion-extension increased from 12.9° in the baseline condition to 13.5° following O1. This subsequently increased to 15.0° following O2 and 15.9° following O3. Lateral bending increased from 16.5° to 17.6° after O1, 18.2° after O2, and 18.7° after O3, Axial rotation increased from 33.8° to 36.3° after O1, 38.1° after O2, and 39.4° after O3.

This increase in ROM translated to a total increase of 4.7% in flexion-extension after O1, 16.3% after O2, and 23.3% after O3 relative to baseline. Lateral bending increased by 6.7%, 10.3%, and 13.3%, after O1, O2 and O3, respectively. Axial rotation increased by 7.4%, 12.7%, and 16.6%, after O1, O2, and O3, respectively (Table 2 and Table 3, Figure 5).

### 3.3. Middle Thoracic Segment T5–T8 from the Single Thoracic Specimen

Flexion-extension increased from 18.7° in the baseline condition to 18.8° following O1. This increased to 21.7° following O2 and 22.6° following O3. Lateral bending increased from 5° to 5.7° after O1, 6.9° after O2, and 8.2° after O3. Axial rotation increased from 23.4° to 26° after O1, 26.3° after O2, and 26.6° after O3 (Table 2 and Table 3, Figure 6).

The highest percent change in ROM was seen in the middle thoracic segment (T5–T8); although flexion-extension initially increased by only 0.5% after O1, this subsequently increased by 16% and 20.9% after O2 and O3, respectively. Lateral bending increased by 14% after O1, 38% after O2, and 64% after O3. Axial rotation increased by 11.1%, 12.4%, and 13.7 after O1, O2 and O3, respectively. 

### 3.4. Lower Thoracic Segment T9–T12 from the Single Thoracic Specimen

Flexion-extension increased from 9.5° at baseline to 10° following O1. This increased to 10.7° following O2 and 10.9° following O3. Lateral bending increased from 9° to 9.2° after O1, 9.6° after O2, and 9.7° after O3. Axial rotation increased from 19° to 19.6° after O1, 20.1° after O2, and 20.2° after O3 (Table 2 and Table 3, Figure 7).

Lastly, the lower thoracic segment (T9–T12) saw the smallest increase in flexion-extension, lateral bending, and axial rotation. From baseline to O1, to O2, and to O3, flexion-extension increased by 5.3%, 12.6%, and 14.7%, respectively. Lateral bending increased by 2.2%, 6.7%, and 7.8%. Axial rotation increased by 3.2%, 5.8%, and 6.3% after O1, O2 and O3, respectively.

## 4. Discussion

Modern surgical treatment of AIS involves deformity correction in multiple planes through translation, derotation, and lengthening of the posterior column to restore kyphosis. The release of posterior elements increases spinal flexibility to allow better deformity correction in multiple planes [24,25].

With a single cadaveric thoracic spine, we demonstrated the feasibility of a novel sequential approach to the Ponte osteotomy, which resulted in an incrementally increasing range of motion in spinal flexion/extension, axial rotation, and lateral bending. SAP osteotomy alone provided a 3.5%, 7.6%, and 7.2% increase in flexion/extension, lateral bending, and axial rotation, respectively, while in situ SP osteotomy provided a 15%, 18%, and 10.3% increase. After these two in situ osteotomies, the spinal canal was still largely protected by the posterior elements. Adding a complete posterior column osteotomy improved flexion/extension, lateral bending, and axial rotation by 19.6%, 28.3%, and 12.2%, respectively. 

The first two osteotomies described (O1, O2) could be performed with an ultrasonic bone-cutting device safely without passing rongeurs through the epidural space. According to our data, they provided roughly 75% of the flexibility gained by a formal Ponte osteotomy (70% in flexion-extension, 77% in lateral bending, and 77% in axial rotation). This biomechanical data supports a stepwise approach to the Ponte osteotomy and demonstrates that stepwise gains in mobility can be achieved while limiting spinal cord exposure. 

When comparing the results from our single cadaveric specimen to the literature, our results are generally comparable in terms of flexion-extension and axial rotation. With lateral bending, however, we found a greater change from baseline than in several other comparable studies.

Holewijn et al. performed a stepwise posterior osteotomy study involving resection of the supra/interspinous ligament (SIL), inferior facet, flaval ligament, superior facet, and rib heads. The authors found an incremental increase in spinal flexibility with diminishing returns after each step [24]. In their study, SIL resection, flaval ligament, and complete facetectomies increased ROM by 29.6% in flexion, 12.1% in extension, 5.5% in lateral bending, and 15.3% in axial rotation [24]. Compared to Holewijn et al., we observed greater lateral bending and axial rotation with an in situ SP osteotomy, which effectively detached the ligamentum flavum without uncovering the underlying spinal canal posteriorly. 

A study by Sangiorgio et al. found that a complete Ponte osteotomy increased flexion by 69%, extension by 56%, and axial rotation by 34%, but only minimally increased lateral bending by 2% [25]. A report by Wang et al. found that a Ponte osteotomy increased flexion by 23%, extension by 15%, and axial rotation by 21%, but only minimally increased lateral bending by 2% [26]. While our single thoracic cadaver showed similar flexion/extension and axial rotation improvements to Wang et al. and less than Sangiorgio et al., we saw a six-times greater lateral bending motion with the complete Ponte osteotomy. In another analogous study, Borkowski et al. used a two-step modification of the Ponte osteotomy in 10 thoracic cadaveric specimens mounted as a large unit from T1–T12 [27]. The authors used a biomechanical testing setup that recorded single plane motion (flexion-extension, lateral bending, axial rotation) after bilateral total facetectomies and increasing numbers of Ponte osteotomies up to four levels. Like Sangiorgio et al. and Wang et al., Borkowski et al. found lateral bending changed less than flexion-extension and axial rotation with a 9% increase from baseline after four-level Ponte osteotomy [27].

One issue in comparing our report to the previous studies is that each study used an intact spine as its baseline reference. We chose to begin from a baseline after SIL resection and inferior facetectomy because these are generally accepted as standard steps of the exposure for posterior spinal fusion. It is unclear why we saw greater lateral bending and coronal flexibility changes, but this highlights the potential variability between cadaveric specimen stiffness, which will be lessened with a greater sample size.

A significant limitation of our technical note was our use of a single adult cadaver specimen. This report is a conceptual demonstration, and greater statistical power was needed to support the proposed technique. Because of our experimental setup, we could not isolate the effect of each osteotomy on flexion versus extension range of motion. However, it was reasonable to assume that the bony resection associated with a formal Ponte osteotomy would better facilitate segmental extension through posterior column shortening. Further study is warranted to investigate this hypothesis. Another limitation of this report was that it was performed on a spinal specimen without deformity. Stiffness varied across patients with scoliosis, and the thoracic hypokyphosis or lordosis that is commonly seen in the thoracic spine of patients AIS may result in posterior spinal ligaments that are more stiff and contracted [28]. Posterior releases in different types and severities of scoliotic curves thus may have variable efficacy depending on rib cage deformation, Cobb angle, sagittal and coronal alignment, and vertebral axial rotation. We designed our experimental setup and loading parameters based on existing literature, but it is possible that the range of motion achieved by applying forces to the end vertebra was not directly correlated to the deformity correction that can be achieved when force is applied during surgery through segmental pedicle screws. 

Another limitation of our technical note is that the range of motion increases seen from our cadaveric spine may not directly correlate with greater deformity correction achievable in vivo. Few studies have reported the forces necessary to achieve deformity correction during spinal surgery, and thus it is difficult to know how our results translate to clinical practice [29,30]. However, this limitation exists in all biomechanical cadaveric studies on spinal destabilization procedures with simulated loads. The load applied in this report of 5 Nm in each plane falls within the range of 2–6 Nm used in other related studies [24,25,26,27,31,32]. 

It should also be noted that the cadaveric model was stripped of stabilizing paraspinal musculature and separated from the anterior chest wall. A study by Mannen et al. using thoracic cadaver specimens with full rib cages found a significant yet small (<1°/Ponte osteotomy) correction in flexion but no significant axial rotation or lateral bending [31]. In our report, specimen preparation was in line with other studies in the literature but may have demonstrated a greater range of motion increases compared to specimens with a full rib cage or the in vivo setting [27,28,32,33]. 

Despite these known limitations, our experimental setup represents the best established and feasible cadaveric method to study posterior releases in an in vitro setting. The results from this technical report align with available retrospective clinical studies [12,13,14].

In vivo, our osteotomy techniques are currently performed freehand without any assistance from advanced technology to facilitate bony resections. However, advances in robotics, computer navigation (NAV), and virtual reality (VR) may one day further improve the execution of spinal corrective osteotomies in AIS [34,35]. VR and NAV have been commonly employed to aid in pedicle screw placement, but less has been published about the use of this technology in performing corrective osteotomies. In one report by Kosterhon et al., they preoperatively created a virtual resection plan for a pedicle subtraction osteotomy. They exported the 3-D plan into a navigation system that could display the planned resection intraoperatively via the surgical microscope’s head-up display [36]. While the authors found the intraoperative visualization helpful, they noted that it might be more relevant in patients undergoing large complex osteotomies, such as a pedicle subtraction osteotomy for hemivertebrae. Our sequential osteotomy technique uses smaller bony resections, and we performed intraoperative manual spinal flexibility testing periodically to titrate the number of vertebral levels included and the degree of posterior release; thus, VR and NAV osteotomy planning appear less applicable for our proposed method at this time. However, it is reasonable to expect that these technologies will continue to offer new opportunities to improve the surgical treatment of AIS. 

## 5. Conclusions

Posterior column osteotomies are safe and effective for the multiplanar correction of adolescent idiopathic scoliosis [21]. This report demonstrates a novel posterior spinal osteotomy sequence to progressively improve flexibility while protecting the spinal canal. Our results suggest that this stepwise risk-minimizing approach of the Ponte osteotomy may be adequate to achieve desired deformity correction in many scenarios and align with our clinical experience using this technique. Complete formal Ponte osteotomy can thus be reserved for severe cases or cases in which posterior column compression is necessary for deformity correction. Further cadaveric and clinical studies are needed to confirm our results.

## Figures and Tables

**Figure 1 children-10-00470-f001:**
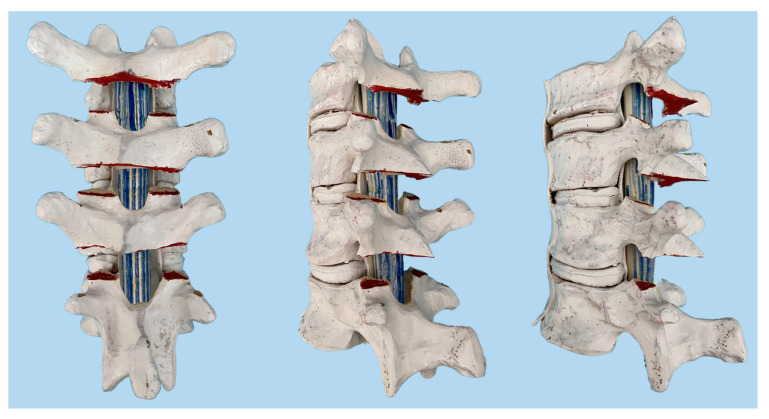
The Ponte osteotomy, as described by Ponte in the thoracic spine.

**Figure 2 children-10-00470-f002:**
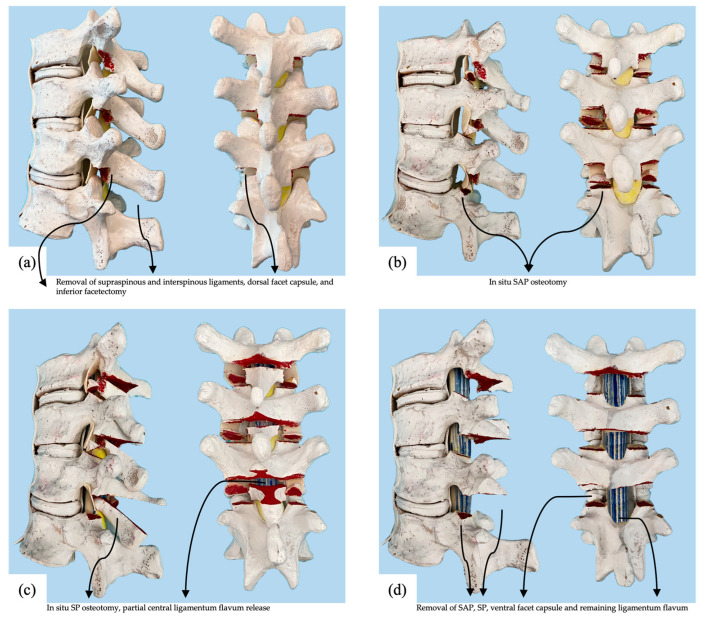
Sawbone thoracic spine models illustrating the experimental conditions of (**a**) baseline, (**b**) SAP osteotomies in situ, (**c**) SP osteotomy in situ, and (**d**) complete posterior column osteotomy.

**Figure 3 children-10-00470-f003:**
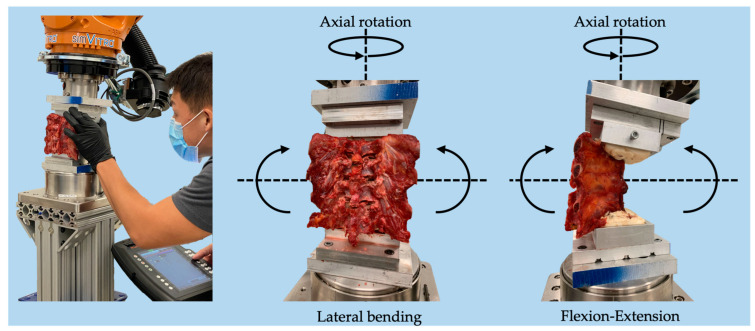
Illustration of experimental setup and directional loading of thoracic sections.

**Figure 4 children-10-00470-f004:**
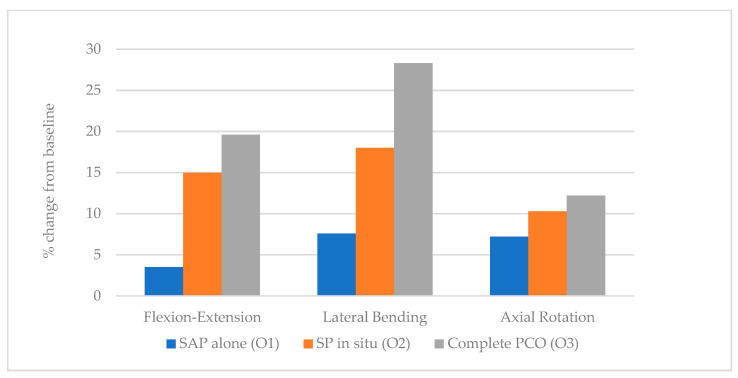
T1–T12 percent change from baseline in response to 5 Nm torque following stepwise osteotomies.

**Figure 5 children-10-00470-f005:**
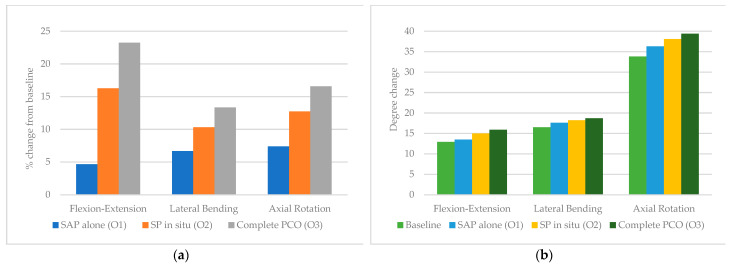
T1–T4 (**a**) percent from baseline and (**b**) degree change in response to 5 Nm torque following stepwise osteotomies.

**Figure 6 children-10-00470-f006:**
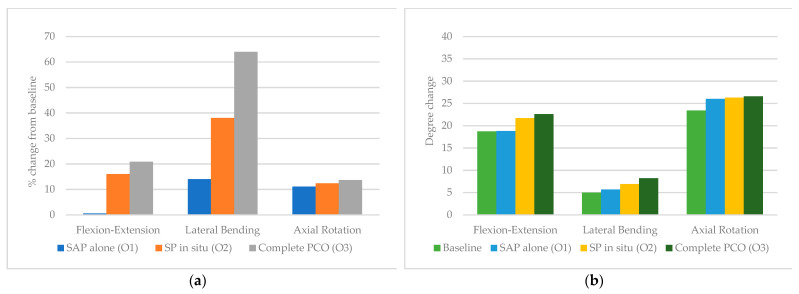
T5–T8 (**a**) percent from baseline and (**b**) degree change in response to 5 Nm torque following stepwise osteotomies.

**Figure 7 children-10-00470-f007:**
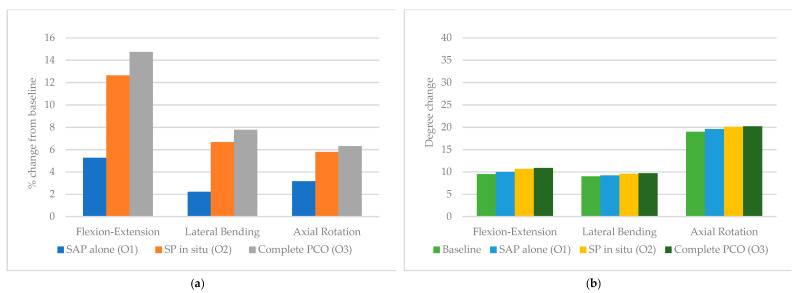
T9–T12 (**a**) percent from baseline and (**b**) degree change in response to 5 Nm torque following stepwise osteotomies.

**Table 1 children-10-00470-t001:** T1–T12 percent change from baseline in response to 5 Nm torque following stepwise osteotomies (% + SD).

T1–T12	SAP Alone (O1)	SAP + SP (O2)	SAP +SP + PC + LF (O3)
Flexion-Extension	3.5 ± 2.6	15.0 ± 2.0	19.6 ± 4.4
Lateral Bending	7.6 ± 5.9	18.3 ± 17.1	28.4 ± 31.0
Axial Rotation	7.2 ± 3.0	10.3 ± 3.9	12.2 ± 5.3

SAP, Superior Articular Process; SP, Spinous Process; PC, Posterior Column; LF, Ligamentum Flavum; O1, 1st osteotomy; O2, 2nd osteotomy; O3, 3rd Osteotomy.

**Table 2 children-10-00470-t002:** Percent change from baseline in response to 5 Nm torque following stepwise osteotomies in each thoracic segment (%).

		SAP Alone (O1)	SAP + SP (O2)	SAP + SP + PC + LF (O3)
T1–T4	Flexion-Extension	4.7	16.3	23.3
	Lateral Bending	6.7	10.3	13.3
	Axial Rotation	7.4	12.7	16.6
T5–T8	Flexion-Extension	0.5	16	20.9
	Lateral Bending	14	38	64
	Axial Rotation	11.1	12.4	13.7
T9–T12	Flexion-Extension	5.3	12.6	14.7
	Lateral Bending	2.2	6.7	7.8
	Axial Rotation	3.2	5.8	6.3

**Table 3 children-10-00470-t003:** Degree change in range of motion in response to 5 Nm torque following stepwise osteotomies (°).

		Baseline	SAP Alone (O1)	SAP + SP (O2)	SAP + SP + PC + LF (O3)
T1–T4	Flexion-Extension	12.9	13.5	15	15.9
	Lateral Bending	16.5	17.6	18.2	18.7
	Axial Rotation	33.8	36.3	38.1	39.4
T5–T8	Flexion-Extension	18.7	18.8	21.7	22.6
	Lateral Bending	5	5.7	6.9	8.2
	Axial Rotation	23.4	26	26.3	26.6
T9–T12	Flexion-Extension	9.5	10	10.7	10.9
	Lateral Bending	9	9.2	9.6	9.7
	Axial Rotation	19	19.6	20.1	20.2

## Data Availability

Data is contained within the article.

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
