# Peer review of "Introduction of a Novel Sequential Approach to the Ponte Osteotomy to Minimize Spinal Canal Exposure"

_children, 2023, doi:10.3390/children10030470_

Round 1

Reviewer 1 Report

This is an excellent biomechanical study which shows the impact of sequential posterior releases of the thoracic spine on flexibility in different planes.  It takes into account newer technology in the form of ultrasonic scalpel which has changed the technique by which these osteotomies can be performed.  This will be of interest to spinal deformity surgeons who must weigh the risk of performing more aggressive releases against the potential benefit in terms of flexibility for correction. 

Author Response

Thank you for your feedback!

Reviewer 2 Report

Thank you for inviting me to review the manuscript entitled "A Stepwise Ponte Osteotomy: Biomechanical Cadaveric Pilot Study of a Risk Minimizing Approach". The authors established an biomechanical model by human cadaveric non-scoliotic thoracic spinal specimen to  study a novel sequential approach to the Ponte osteotomy with minimizing spinal canal exposure. The manuscript needs some modifications before publication.

1)Line 43. The content about risk minimizing approach should be described in more detail in introduction section.

2)Line 58. The authors mentioned "One fresh-frozen human cadaveric thoracic spine specimen". Donor's age should be reported. It is not appropriate to study AIS with adult or elderly bones. This cadaveric thoracic spine specimen is AIS spine?

3)Line 58. Only ONE specimen can effectively support the conclusion?

4)Line 88. What pathophysiological conditions were simulated by authors' loading conditions? stand? prostrate? Is this loading condition validated?

5)Line 230. The authors must declare that the current model wae created based on the skeletal system, without considering the effect of paraspinal muscle and thoracic structure, which significantly affects the stability of thoracic spine. 

Author Response

1)Line 43. The content about risk minimizing approach should be described in more detail in introduction section.

Response: Added more detail about risk minimizing approach in introduction.

2)Line 58. The authors mentioned "One fresh-frozen human cadaveric thoracic spine specimen". Donor's age should be reported. It is not appropriate to study AIS with adult or elderly bones. This cadaveric thoracic spine specimen is AIS spine?

Response: Added age and BMI of specimen. Currently it is not feasible to obtain adolescent cadaveric specimens, including those with untreated deformities, thus cadaveric specimens from adult donors are used. This is an established model to study AIS despite the doners being adult. This model is well documented in the literature. A few examples:

  • Borkowski, Sean L., et al. "Flexibility of thoracic spines under simultaneous multi-planar loading." European Spine Journal26 (2017): 173-180.
  • Mannen, Erin M., et al. "Influence of sequential Ponte osteotomies on the human thoracic spine with a rib cage." Spine Deformity5 (2017): 91-96.
  • Holewijn, Roderick M., et al. "How does spinal release and ponte osteotomy improve spinal flexibility? The law of diminishing returns." Spine Deformity 3.5 (2015): 489-495.

3)Line 58. Only ONE specimen can effectively support the conclusion?

Response: This study is a demonstration of feasibility. Our conclusion is that this osteotomy shows promise as a technique that will sequentially increase spinal flexibility. Because we were able to show consistent results across the three thoracic specimens, we believe our results are valid and set the stage for a larger study with many more cadaveric specimens.

4)Line 88. What pathophysiological conditions were simulated by authors' loading conditions? stand? prostrate? Is this loading condition validated?

Response: The loading conditions applied to the specimens are standard measures used in the literature to evaluate overall multiplanar spinal flexibility. The goal of this study is evaluate how our osteotomy technique changes spinal flexibility in different planes of motion (flexion/extension, lateral bending, axial rotation). Increasing spinal flexibility in multiple planes allows surgeons to better improve scoliotic deformity with the final fusion construct. For other instances of these loading conditions in the literature, here are a few examples:

  • Borkowski, Sean L., et al. "Flexibility of thoracic spines under simultaneous multi-planar loading." European Spine Journal 26 (2017): 173-180.
  • Anderson, Andy L., et al. "The effect of posterior thoracic spine anatomical structures on motion segment flexion stiffness." Spine 34.5 (2009): 441-446.
  • Wang, Cheng, et al. "Biomechanical comparison of ponte osteotomy and discectomy." Spine 40.3 (2015): E141-E145.

5)Line 230. The authors must declare that the current model wae created based on the skeletal system, without considering the effect of paraspinal muscle and thoracic structure, which significantly affects the stability of thoracic spine.

Response: Added a clarification about the experimental setup in the Methods section.

Reviewer 3 Report

Authors present a biomehanical cadaver pilot study of a risk minimizing approach for Ponte osteotomy -with removal of spinous process as the last step and minimal opening of the spinal canal - on three sections of thoracic spine;  each segment was loaded with 5Nm in four different conditions; ligamentum flavum release/PCO increased flexion/extension, lateral bending, and axial rotation most compared to other three conditions by 19.6%, 28.3%, and 12.2%, respectively, suggesting that incremental increases in range of motion can be achieved with minimal spinal canal exposure.

Figures which show the "intraoperative" situation and certain steps should be enlarged and more clear as well as marked. We also suggest to include a scheme, photo or a video of robotic simulation of movements for better unterstanding. Since Ponte procedure in patients with adolescent idiopathic scoliosis is usually used for correction of thoracolumbar deformities, a choice of solely thoracic cadaver is not the best one - especially since the range of movement in a complete thoracic-lumbar setting different is than in the thoracic spine alone - this should be stated in the Limitations. 

We also suggest to include the osteotomy video; otherwise, the manuscript is understandable only to a small community of spine surgeons who perform this procedure. Authors need to explain what is the actual novelty in this procedure and what does a "classical" Ponte procedure means for them - and what is here new; classical Ponte procedure needs to be explained in detail. On the Figure under D there is a spinous process still left (partialy or complete) so this requires further explanation. 

Furthermore, I would not call this cadaver experiment a study, but a technical note. 

Major drawback is that there is a single cadaver and that the same experiments should have been performed on a further cadaver with the classical Ponte procedure - in this way we would know if there is really any advantage of this novel approach. 

Important literature is missing and needs to be discussed:

Wang C, Bell K, McClincy M, Jacobs L, Dede O, Roach J, Bosch P. Biomechanical comparison of ponte osteotomy and discectomy. Spine (Phila Pa 1976). 2015 Feb 1;40(3):E141-5. doi: 10.1097/BRS.0000000000000697. PMID: 25384049. - please include and discuss, here we also see the importance of comparison between two methods; furthermore, I suggest to comment on effects on coronar deformity correction 

Mannen EM, Arnold PM, Anderson JT, Friis EA. Influence of Sequential Ponte Osteotomies on the Human Thoracic Spine With a Rib Cage. Spine Deform. 2017 Mar;5(2):91-96. doi: 10.1016/j.jspd.2016.10.004. PMID: 28259271.

Role of virtual reality, neuronavigation and robotics - navigated osteotomies - should be included into Discussion.

Author Response

Figures which show the "intraoperative" situation and certain steps should be enlarged and more clear as well as marked. We also suggest to include a scheme, photo or a video of robotic simulation of movements for better unterstanding. Since Ponte procedure in patients with adolescent idiopathic scoliosis is usually used for correction of thoracolumbar deformities, a choice of solely thoracic cadaver is not the best one - especially since the range of movement in a complete thoracic-lumbar setting different is than in the thoracic spine alone - this should be stated in the Limitations.

Response: Edited figures to increase size and add labels. Added figure of the experimental setup. Based on the literature, we find the Ponte osteotomy in many AIS patients involves major thoracic curve correction, and most similar cadaveric studies focus on thoracic correction. We did add to the introduction that some authors (Shufflebarger et al.) have described the Ponte osteotomy for lumbar and thoracolumbar curves. A sample of supporting literature is below:

  • Holewijn, R.M.; Schlösser, T.P.; Bisschop, A.; Van Der Veen, A.J.; Stadhouder, A.; Van Royen, B.J.; Castelein, R.M.; De Kleuver, M. How does spinal release and ponte osteotomy improve spinal flexibility? The law of diminishing returns. Spine Deformity 2015, 3, 489-495.
  • Shah, S.A.; Dhawale, A.A.; Oda, J.E.; Yorgova, P.; Neiss, G.I.; Holmes, L.; Gabos, P.G. Ponte osteotomies with pedicle screw instrumentation in the treatment of adolescent idiopathic scoliosis. Spine deformity 2013, 1, 196-204.
  • Borkowski, Sean L., et al. "Flexibility of thoracic spines under simultaneous multi-planar loading." European Spine Journal 26 (2017): 173-180.
  • Shufflebarger, Harry L., Matthew J. Geck, and Cynthia E. Clark. "The posterior approach for lumbar and thoracolumbar adolescent idiopathic scoliosis: posterior shortening and pedicle screws." Spine 29.3 (2004): 269-276.

We also suggest to include the osteotomy video; otherwise, the manuscript is understandable only to a small community of spine surgeons who perform this procedure. Authors need to explain what is the actual novelty in this procedure and what does a "classical" Ponte procedure means for them - and what is here new; classical Ponte procedure needs to be explained in detail. On the Figure under D there is a spinous process still left (partialy or complete) so this requires further explanation.

Response: We are currently working on a video for the described technique, but are unable to provide in the short period of 10 days for these reviews. It is an active project that we will hopefully be able to provide soon. We have expanded the introduction to include a very detailed description of the Ponte osteotomy by Dr. Ponte and how our technique differs. We have edited Figure 3 (D) to remove the residual spinous process to avoid confusion.

Furthermore, I would not call this cadaver experiment a study, but a technical note.

Response: We have changed areas in the text describing a study to a technical note.

Major drawback is that there is a single cadaver and that the same experiments should have been performed on a further cadaver with the classical Ponte procedure - in this way we would know if there is really any advantage of this novel approach.

Response: We do agree that our next goal is to obtain more cadaveric specimens to repeat the experiment.  

Important literature is missing and needs to be discussed:

Wang C, Bell K, McClincy M, Jacobs L, Dede O, Roach J, Bosch P. Biomechanical comparison of ponte osteotomy and discectomy. Spine (Phila Pa 1976). 2015 Feb 1;40(3):E141-5. doi: 10.1097/BRS.0000000000000697. PMID: 25384049. - please include and discuss, here we also see the importance of comparison between two methods; furthermore, I suggest to comment on effects on coronar deformity correction

Response: Added to discussion, and discussed similar smaller coronal correction seen in study as with Sangiorgio et al. (Sangiorgio, et al. Quantification of increase in three-dimensional spine flexibility following sequential Ponte osteotomies in a cadaveric model. Spine Deformity 2013, 1, 171-178.)

Mannen EM, Arnold PM, Anderson JT, Friis EA. Influence of Sequential Ponte Osteotomies on the Human Thoracic Spine With a Rib Cage. Spine Deform. 2017 Mar;5(2):91-96. doi: 10.1016/j.jspd.2016.10.004. PMID: 28259271.

Response: Added citation and discussion.

Role of virtual reality, neuronavigation and robotics - navigated osteotomies - should be included into Discussion.

Response: Added to discussion.

Reviewer 4 Report

Dear Authors,

Congratulations for such a paper and your research.

Described novel posterior spinal osteotomy sequence was very well described as a technique improving flexibility while protecting spinal canal. First of all, posterior column osteotomies were also very well described in Introduction section. Methodology was very clear presented as well as Results. Results of your research were properly discussed regarding references that were shown at the section References.

Good luck with your future researches and best regards,

FM

Author Response

Thank you for your comments!

Reviewer 5 Report

This manuscript is well written and introduces a new surgical technique to the community. However, even with the qualification of "pilot study" in the title, is a demonstration of feasibility, not a biomechanical study. 

Title: The title should be changed to "Introduction of a novel sequential approach to the Ponte Osteotomy to Minimize Spinal Canal Exposure." 

Abstract: Modify to reflect the scope of the manuscript as an introduction of a technique, not a biomechanical study.

Lines 14-15: Suggest rephrasing with "we introduce a novel sequential approach" and delete "We use a cadaveric model to pilot a.."

Lines 25-26: Delete "our results suggest..." and replace with "Our study introduced a novel approach... and demonstrated feasibility in a cadaveric model."

Introduction:

Lines 46-55: Are the same bony and ligamentous structures ultimately resected for the novel method, as compared to a traditional Ponte osteotomy? Is it the method/technique the only thing that differs, or are there any ligaments that remain attached for the novel method, which would be resected using a traditional Ponte?

Materials and Methods: 

This section should be reorganized to emphasize the surgical procedure, as opposed to the biomechanical testing. As written, the emphasis is the study design, not the novel surgical technique. The methods described are not innovative, nor was a sufficient sample size included.

Results: As presented, the results suggest that multiple samples were included, and that this was a complete study. This is misleading, as only one thoracic spine, dissected into three specimens, was included. This is a significant limitation for a study, which is why the "biomechanical study" portion should be minimized, and presented as a demonstration of feasibility.

Discussion: 

Please review Borkowski, Sangiorgio, Scaduto et al. ESJ 2017, for a highly related study that should be added and acknowledged in your discussion.

Author Response

Title: The title should be changed to "Introduction of a novel sequential approach to the Ponte Osteotomy to Minimize Spinal Canal Exposure."

Response: Changed title

Abstract: Modify to reflect the scope of the manuscript as an introduction of a technique, not a biomechanical study.

Response: Modified abstract.

Lines 14-15: Suggest rephrasing with "we introduce a novel sequential approach" and delete "We use a cadaveric model to pilot a.."

Response: Edited.

Lines 25-26: Delete "our results suggest..." and replace with "Our study introduced a novel approach... and demonstrated feasibility in a cadaveric model."

Response: Edited.

Introduction:

Lines 46-55: Are the same bony and ligamentous structures ultimately resected for the novel method, as compared to a traditional Ponte osteotomy? Is it the method/technique the only thing that differs, or are there any ligaments that remain attached for the novel method, which would be resected using a traditional Ponte?

Response: We clarified in the introduction that our sequential method ends in the same manner as the Ponte osteotomy. 

Materials and Methods:

This section should be reorganized to emphasize the surgical procedure, as opposed to the biomechanical testing. As written, the emphasis is the study design, not the novel surgical technique. The methods described are not innovative, nor was a sufficient sample size included.

Response: Reorganized methods to emphasize technique.

Results: As presented, the results suggest that multiple samples were included, and that this was a complete study. This is misleading, as only one thoracic spine, dissected into three specimens, was included. This is a significant limitation for a study, which is why the "biomechanical study" portion should be minimized, and presented as a demonstration of feasibility.

Response: Reworked the results section to focus on the single cadaveric spine specimen.

Discussion:

Please review Borkowski, Sangiorgio, Scaduto et al. ESJ 2017, for a highly related study that should be added and acknowledged in your discussion.

Response: Added to discussion.

Round 2

Reviewer 3 Report

Authors have answered some of the reviewer remarks. I suggest to oblige the authors to provide a video of the procedure as a supplement in the following months.